# Synthesis of Hydroxyapatite/Bioglass Composite Nanopowder Using Design of Experiments

**DOI:** 10.3390/nano12132264

**Published:** 2022-06-30

**Authors:** Shamsi Ebrahimi, Coswald Stephen Sipaut

**Affiliations:** Faculty of Engineering, Universiti Malaysia Sabah, UMS Road, Kota Kinabalu 88400, Malaysia; ebrahimi_shamsi63@yahoo.com

**Keywords:** composite, bioglass, hydrothermal, combined design

## Abstract

Composite scaffolds of hydroxyapatite (HAp) nanoparticles and bioactive glass (BG) were applied as an appropriate selection for bone tissue engineering. To this end, HAp/BG composite was synthesized by a hydrothermal method using Design of Experiments (DOE) with a combined mixture–process factor design for the first time. The input variables were hydrothermal temperature at three levels (i.e., 100, 140, 180 °C) as a process factor and two mixture components in three ratios (i.e., HAp 90, 70, 50; BG 50, 30, 10). The degree of crystallinity and crystal size in the composite were the output variables. XRD showed that only a small fraction of BG was crystallized and that a wollastonite phase was produced. The XRD results also revealed that incorporation of Si into the HAp structure inhibited HAp crystal growth and restricted its crystallization. The FTIR results also showed that the intensity of the hydroxyl peak decreased with the addition of silicon into the HAp structure. DOE results showed that the weight ratio of the components strongly influenced the crystal size and crystallinity. SEM and FTIR results identified the greatest bioactivity and apatite layer formation in the Si-HAp sample with an HAp70/BG30 ratio after 14 days immersion in simulated body fluid (SBF) solution, as compared to other ratios and HAp alone. Therefore, the combination of HAp and BG was able to yield a HAp/BG composite with significant bioactivity.

## 1. Introduction

Tissue engineering uses a combination of engineering techniques, material science, and biochemical and physiochemical knowledge to replace biological tissue. Materials with calcium phosphate are ideal for implantation into bone because of their similarity in chemical composition to bone tissue. In recent years, a wide range of calcium phosphate ceramics have been used in the repair and reconstruction of bone. Among them, nano hydroxyapatite (HAp) has received considerable attention since it is a main component of the mineral part of extracellular matrix. The most important HAp features include bioactivity and non-toxicity; the bioactivity of HAp enables it to directly bond with the body cells [1,2]. Moreover, the presence of this material is also able to differentiate mesenchymal stem cells from osteoblast cells or other bone cells [3].

Particle size plays a key role in the biological behavior of HAp. In this regard, nano synthesis of HAp powders with a high specific surface has been the basis of research studies. Nano-sized particles of HAp ceramics will perform their biomaterial functions better and have higher strength [4,5] and compared to their similar micro-particles, they have much better properties. In fact, using nano-sized HAp particles instead of micrometric particles of HAp has been very common due to its high specific area, better biodegradability, and higher bioactivity. Because of the structural similarity of these nanoparticles to real bone samples, the probability of rejection of implantation in the body is significantly reduced [6,7]. Moreover, nanosized particles absorb more protein, which increases adhesion and cell proliferation. The ability to move into cells of HAp nanoparticles can be used for therapeutic delivery of a wide range of proteins, antibodies, oligonucleotides, imaging agents, and liposomes in various biological conditions and systems [8]. In addition, in a study by [9], it was shown that at the same size of HAp nanoparticles, adhesion, proliferation, and osteoblast differentiation on bone marrow mesenchymal cells of HAp with a high degree of crystallinity were more than amorphous calcium phosphate. This research showed the crystallinity grade of HAp could be an effective factor in its bioactivity [9].

Despite the high biocompatibility of HAp, its bioactivity can be increased using different methods. For this reason, many attempts in recent years have been made to combine HAp with more active bioceramics and biological glasses to obtain a new generation of composite biomaterials with appropriate biological properties [10]. Composite scaffolds of HAp nanoparticles and bioactive glass (BG) have been applied as an adequate selection for bone tissue engineering applications [11]. Bioactive glass is one biological material capable of chemical binding with hard and soft tissue, and bioactive glasses at a nano scale are more biocompatible due to nano-size pores and increased surface area, which enhance dissolution rate and accelerate biocompatibility mechanisms [12,13]. A significant feature of silicate biomaterial is its ability to release silicon ions at a concentration that improves the growth and differentiation of osteoblast cells [14].

Design of Experiments (DOE) is considered an important tool for improving the functionality of a production process and is also widely used in the development of new processes. One of the objectives of DOE is process optimization to achieve desired conditions. In order to achieve this goal, surface response methods have been used to design tests. Response surface methodology (RSM) is a modern and effective technique for the development, improvement, and optimization of processes that can simultaneously analyze multiple variables and optimize their response. This technique is effective in the design, development, and formulation of new products and in improving the quality of existing products [15].

Therefore, the objective of this study was to synthesize and characterize HAp/BG composite nanopowder via the hydrothermal method by using a design experiment. The effects of hydrothermal temperature and component ratio on crystal size and crystallinity were investigated. To this end, the combined mixture process design of DOE was used to synthesize an HAp/BG composite and total 11 experiments were designed in order to find the appropriate proportions of the HAp and glass phases and the appropriate hydrothermal temperature that will achieve high crystallinity and the best crystal size of HAp/BG composite as responses, for the first time. A quadratic model for mixture components (i.e., HAp and BG) and a linear model for process factors (i.e., temperature) were developed to investigate the effect of experimental variables and their interaction on HAp/BG crystallinity and crystal size. Finally, the bioactivity behavior of HAp and HAp/BG powders was investigated with immersion in simulated body solution (SBF) for 14 days.

## 2. Materials and Methods

### 2.1. Synthesis of HAp

HAp was synthesized based on our previous paper [16] with a hydrothermal method using two raw materials (Merck, Darmstad, Germany): Ca(NO_3_)_2_.4H_2_O as calcium precursor and (NH4)_2_HPO_4_ as phosphate precursor (Equation (1)). In summary, 4.723 g (1 M) of Ca(NO_3_)_2_.4H_2_O and 1.584 g (0.67M) of (NH4)_2_HPO_4_ were dissolved in 20 mL of deionized water. White suspension of HAp was obtained by adding a solution of diammonium phosphate to a solution of calcium nitrate tetrahydrate under continuous stirring for 1 h. NH_3_ (Merck, Darmstadt, Germany) was used to adjust the pH of the solution to 10. Solutions prepared with pH = 10 were hydrothermally treated at 130 °C for 10 h. The final solution was filtered and washed with a mixture of ethanol and distilled water (volume ratio of 1:1) and then dried in an oven for 10 h at 60 °C.
(1)$$10\mathrm{Ca}{\left({\mathrm{NO}}_{3}\right)}_{2}\cdot 4H_{2}O+6{\left({\mathrm{NH}}_{4}\right)}_{2}{\mathrm{HPO}}_{4}+8{\mathrm{NH}}_{4}\mathrm{OH}\to {\mathrm{Ca}}_{10}{\left({\mathrm{PO}}_{4}\right)}_{6}{\left(\mathrm{OH}\right)}_{2}+20{\mathrm{NH}}_{4}{\mathrm{NO}}_{3}+46H_{2}O$$

### 2.2. Synthesis of BG (77 s)

The glass samples were prepared by acid catalyzed sol-gel method assisted by hydrothermal process. Chemicals with weight composition 80 mol% SiO_2_: 14 mol% CaO: 6 mol% P_2_O_5_ was selected as in the previous study [17]. A brief description: hydrolysis of tetramethyl orthosilicate (TMOS, Sigma-Aldrich, St. Louis, MO, USA) and phosphoric acid (H_3_PO_4_, Sigma-Aldrich, St. Louis, MO, USA) was catalyzed with a solution of 0.1 mol L-1 HNO_3_ (Sigma-Aldrich, St. Louis, MO, USA). Starting with the hydrolysis of TMOS for 45 min, the other reagents were added sequentially at 60 min intervals, under constant stirring. Before reaching the gel point, the sol components were placed in a stainless-steel autoclave lined with a Teflon core and heated in an oven at 250 °C for 24 h. The producing product was dried at 100 °C.

### 2.3. Synthesis of HAp/BG Nanocomposite Powder

HAp/BG powders were synthesized based on the method presented in our previous paper [17]. Briefly, nanocomposite powder was mixed with various proportions of synthesized HAp, and bioactive glass powder was mixed with (SiO_2_, CaO, P_2_O_5_) compound. HAp/BG powder with various weight ratios was prepared based on the combined design, in alkaline conditions (pH = 10) under stirring to make a suitable solution. Distilled water was used as a liquid phase. The final solution was transferred to a 100 mL Teflon container and was hydrothermally treated at different temperatures for 10 h. After hydrothermal treatment, the obtained solution was filtered and then washed with distilled water. The powder was dried in an oven at 100 °C for 10 h and then sintered at 700 °C (Figure 1).

XRD analysis was carried out using an X-ray diffractometer (X’pert Pro, PANalytical BV, Almelo, The Netherlands) with a CuKa radiation source (λ = 1.54056 Å) and operated at 40 kV and 30 mA. The diffraction patterns at room temperature were recorded at 2θ range 10–80°, with a step size of 0.02° and time per scan 1 s.

The average crystallite sizes of obtained powders were calculated using the Debby Scherrer equation (Equation (2)). The degree of crystallinity was also determined using Equation (3).
(2)$$D=\frac{\;K\lambda }{\beta \cos\theta }$$
where D is crystal size, β is the full width at half maximum of the peak (in radians) of the (002) reflection, θ is the Bragg angle in degrees, K is the shape constant (equal to 0.9), and λ is the X-ray wavelengths (λ = 1.5405 Ǻ) [18,19].
(3)$$\mathrm{Crystallinity}\left(\%\right)=\frac{\sum A_{c}}{\sum A_{c}+\sum A_{A}}\times 100$$
where ΣAC + ΣAA gives the sum of the area under all the HAp and HAp/BG crystalline and amorphous peaks, and ΣAC yields the sum of the areas under the crystalline peaks present in the scan range between 10 to 80° [20].

The IR analysis was recorded using Fourier Transform Infrared Spectroscopy (FTIR, Perkin Elmer, Waltham, MA, USA) in the frequency range 400–4000 cm^−1^ at room temperature. The SEM/EDX analysis was obtained using a Scanning Electron Microscope (Hitachi S3400 N, Tokyo, Japan) equipped with EDX (Quantax 200, Bruker, Bremen, Germany) to study and investigate the apatite layer on the HAp and HAp/BG nanopowder after immersion into SBF solution.

### 2.4. Investigation of Sample Bioactivity in SBF Solution

Simulated body fluid (SBF) solution was used for evaluating the bioactivity of samples and was prepared based on the Kokubo guidelines [21,22]. To perform the bioactivity tests, 100 mL glass bottles with caps were utilized, and 1 mL SBF solution was used per 1 mg of sample. Samples were immersed in the SBF solution and incubated under static condition at 37 °C for 14 days. After this time, they were washed with distilled water and dried at an ambient temperature.

### 2.5. Design of Experiments

#### Combined Mixture–Process Design of DOE

Mixed design is another optimization method that is considered a group of RSMs in which the product of interest comprises several components. When a product is formed by mixing two or more components, this is called “mixing” and the elements forming the mixture are called “components”. In this group of designs, a response is a function of the proportions of various components in the mixture. One of the functions of designing mixed tests is to find the best proportion of each component in the mixture as well as the best amount of each variable in the process in order to optimize a single response or multiple responses simultaneously.

When a set of process factors and components of a mixture are simultaneously effective on the response of the process, combination design or combined mixture–process factors are used. In general, there are two strategies for designing combinational tests:Determining the proper formulation for mixed components with mixed design.Optimization of effective process parameters.

When these two strategies are merged, this design system is called a combination design system. For example, if three components (X_1_, X_2_, and X_3_) of a process have three levels (0, 1/2, and 1) and two process factors have two levels (1, −1), the following figure is a combination agent for this simple design (Figure 2). In general, in mixture design with q components, a mixture region of regular geometry has dimensionality q−1. Therefore, the mixture region of a q = 2 component mixture corresponds to a line, a q = 3 component mixture to a triangular region, and a q = 4 component mixture to a tetrahedral region [23,24].

In Figure 1, the vertices of the triangle indicate a pure component of the mixture, and the mid-points or binary points indicate binary mixtures.

The mathematical model for a mixture has the following three components (Equation (4)):(4)$$Y\left(X\right)=\beta_{1}{Χ}_{1}+\beta_{2}{Χ}_{2}+\beta_{3}{Χ}_{3}+\beta_{12}{Χ}_{1}{Χ}_{2}+\beta_{13}{Χ}_{1}{Χ}_{3}+\beta_{23}{Χ}_{2}{Χ}_{3}$$

For a factorial model that considers the interactions, the equation is as shown in Equation (5):(5)$$Y\left(Z\right)=a_{0}+a_{1}z_{1}+a_{2}z_{2}+a_{1}{}_{2}z_{1}z_{2}$$

The equation of a combination model is as shown in Equation (6):(6)$$Y\left(X,Z\right)=Y\left(X\right)\times Y\left(Z\right)$$

As shown above, there are six terms for the mixed model and two terms for the factorial model, so there are 24 terms in total for this mixed design. In other words, there are 24 tests required to control the parameters that control the overall process. In order to design this model, Design Expert software (version 11, 2018, Stat-Ease Inc, Minneapolis, MN, USA) and combination design were used [25].

The combined mixture–process factor DOE was conducted via Design Expert 11 software. In the synthesis of HAp/BG composite, there are two mixture components, (i.e., HAp and BG) with three different ratios (Table 1) and 1 numeric factor (i.e., temperature) with three levels (Table 2). Notably, the process factor level and ratios were selected based on the literature review [26,27,28,29,30,31,32,33,34]. The effects of parameters such as hydrothermal temperature and the effects of the two mixture components were investigated by combined design via D-optimal method. The design summary is shown in Table 3.

## 3. Results and Discussion

### 3.1. XRD Analysis

Figure 3 shows the XRD peaks of pure HAp powder. This combination of diffraction peaks at 2Ө = 25.90 (002), 2Ө = 31.725 (211), 2Ө = 32.18 (112), and 2Ө = 32.86 (300) corresponds to the hexagonal structure of pure HAp (Yadav et al., 2020). Figure 3 also shows the XRD patterns for the synthesized HAp/BG at different ratios and at three different temperatures (100 °C, 140 °C, and 180 °C) after sintering at 700 °C. The results show that two phases were formed, namely HAp and wollastonite (i.e., calcium silicate, CaSiO_3_) structures. The combination of SiO_2_- P_2_O_5_-CaO (from BG) and the sintering process resulted in the formation of wollastonite. Ref [35] reported that the wollastonite phase is formed at high temperatures due to the combination of SiO2 and CaO particles in BG. It can be seen that after sintering HAp/BG composite powder at 700 °C, only a fraction of the glass phase crystallized to wollastonite with the remaining amorphous (i.e., the peaks related to wollastonite are weak). This is also in accordance with the findings by [28], who found that the wollastonite phase was formed by sintering HAp/BG at above 700 °C [28].

From Figure 3, XRD analysis revealed that HAp in the composite powder did not decompose during sintering at 700 °C since carbonate impurities such as alpha-tricalcium phosphate (α-TCP) and β-tricalcium phosphate are not present. Therefore, a suitable temperature for sintering HAp/BG powder was 700 °C. It is also clear from Figure 3 that pure HAp shows higher peak intensity (corresponding to HAp peaks) than the HAp/BG composite.

It was observed that by increasing the hydrothermal temperature from 100 to 180 °C, the intensity of the peaks in the XRD patterns increased and that at higher temperatures, more crystallization took place.

Table 4 shows the crystallinity and crystal size of HAp/BG composite powder at three different hydrothermal process temperatures (100, 140, 180 °C). Table 4 shows that for the samples produced at 180 °C, lower HAp content resulted in lower crystallinity and smaller crystal sizes. Similar behavior was found for the other temperatures, suggesting that the HAp/BG ratio is an important factor in HAp crystallinity and crystal size. The authors of [36] investigated the effects of incorporating silicon on the degree of crystallinity and crystal size of HAp in HAp/BG composite and found that both responses decreased as a result of the silicon phase in the BG inhibiting HAp crystal growth by limiting atomic arrangement [36]. Research on the incorporation of silicon into the formulation also showed decreased crystallinity and crystal size of HAp in the HAp/BG composite [37].

Table 4 shows that for a fixed HAp/BG ratio in the composite, HAp crystal size increased with greater hydrothermal process temperature (i.e., from 100 to 180 °C). Authors of [38] studied the effects of hydrothermal process temperature on HAp crystal size. Their findings indicate that a higher hydrothermal process temperature promotes crystal growth because it provides higher activity levels for smaller apatite crystals that are bound and grow along the C-axis.

Table 4 also shows that, for a fixed HAp/BG ratio, the degree of crystallinity and the intensity of peaks in the HAp phase increased with hydrothermal process temperature. In a study by [39], increased hydrothermal process temperature was found to increase the degree of crystallinity in HAp [39].

### 3.2. FTIR Analysis

Figure 4 shows the FTIR results of HAp and HAp/BG composite nanopowder under different conditions based on DOE. Figure 4 shows the FTIR spectra of pure HAp obtained at pH 10 and 130 °C. The characteristic peaks are at 536, 602 cm^−1^ and the peaks in the region 950–1100 cm^−1^ are related to a phosphate group. The peaks around 633 and 3568 cm^−1^ are related to an OH- group in the HAp structure. In the HAp/BG composite after sintering, a small amount of BG phase was crystallized and a wollastonite phase was formed. In addition, during the sintering of HAp/BG composite at 700 °C, HAp decomposed and BG entered its structure. Therefore, from the FTIR spectra in Figure 4, the wavenumbers in the range 798 cm^−1^ to 461 cm^−1^ are attributed to Si-O bonds of glass in the HAp structure and wollastonite phase which, by increasing the BG ratio, increase the intensity of the Si-O band [40]. As seen in Figure 4, the peaks of the PO_4_^−2^ group in pure HAp (950–1100 cm^−1^) are more intense than the PO_4_^−2^ peaks in the HAp/BG composite, which indicate the incorporation of silicon into the HAp structure [41].

The FTIR results also showed that the intensity of the hydroxyl peak decreased with the addition of silicon into the HAp structure. In another study by [34], the substitution of silicate groups for phosphate groups removes OH^−^ groups in order to maintain balance [34,37,42]. According to this process, the chemical formula of Si-HA can be determined as Equation (7) [43].
(7)$$10{\mathrm{Ca}}^{+2}+\left(6-x\right){\mathrm{PO}}_{4}^{-3}+{\mathrm{xSiO}}_{4}^{-4}+\left(2-x\right){\mathrm{OH}}^{-}\to {\mathrm{Ca}}_{10}{\left({\mathrm{PO}}_{4}\right)}_{6-x}{\left({\mathrm{SiO}}_{4}\right)}_{x}{\left(\mathrm{OH}\right)}_{\left(2-x\right)}\;\mathrm{with}\;0\leq x\leq 2$$

From Figure 4, increasing hydrothermal temperature at a constant HAp/BG ratio increases the intensity of functional groups and the degree of crystallinity. In studies by [38], it was also found that increasing hydrothermal temperature causes an increased degree of crystallinity.

### 3.3. SEM and EDX Analsis

Figure 5 shows SEM images of HAp/BG nanocomposite powders with different weight ratios at 180 °C. It is clear from these images that the morphology of all samples is a tightly packed compact of fine particles, especially notable for HAp70/BG30. It is also seen that, except for HAp50/BG50, all samples were aggregated due to a smaller particle size which tends to agglomerate. It is worth noting that HAp70/BG30 has a smooth and homogeneous structure, which is an important factor in improving mechanical strength [36].

Figure 5 also shows the results of elemental analysis by EDX spectroscopy of the HAp/BG composite nanopowders at 180 °C. This analysis indicates the presence of all the constituent elements (i.e., Ca, P, O, and Si) of HAp/BG composite powder. Since the applied glass contains (P_2_O_5_) and (SiO_2_), the calculation of the Ca/P ratio of HAp is as in Table 5. Therefore, due to the presence of calcium in the glass phase, the calcium to phosphate ratio has increased. Similar results have been found by [44,45]. However, because of the substitution of the silanol group with the phosphate group, the amount of the phosphate group in the HAp structure was lower, leading to an increased Ca/P ratio in HAp/BG. In this study, the Ca/P ratios for HAp50/BG50, HAp70/BG30, and HAp90/BG10 were 1.87, 1.71, and 1.69, respectively. Similar results have been found by [46]. Additionally, from the EDX results shown in Table 5 and Figure 5, C atoms (i.e., Carbon) were not detected in any of the three HAp/BG composite nanopowders, which means that there was no carbonate impurity. Hence, the Si-HA produced is almost pure and the results from EDX and XRD are in perfect accord.

### 3.4. Comparison of DOE and Experimental Results

In this study, a combined mixture–process design was used to optimize one process factor (i.e., hydrothermal reaction temperature at three levels) and two mixture components (i.e., HAp and BG in three different ratios) with respect to the response variables degree of crystallinity and crystal size of HAp/BG composites. Table 6 reports the values for these variables as predicted by DOE versus the actual experimental data.

### 3.5. Analysis of Variance for HAp Crystal Size in HAp/BG Composite Nanopowder

The first step in analysis of variance (ANOVA) is to select an appropriate model for the system that will accurately predict the experimental results. A quadratic model and a linear model were used for mixture order and process order, respectively. Results of ANOVA for model evaluation and meaningfulness are indicated in Table 7 in terms of *p*-value, R^2^, R^2^ adjusted, R^2^ predicted, and adequate precision. The *p*-value of the surface indicates the significance of the produced model and values less than 0.05 (*p* ≤ 0.05) are considered highly significant. From Table 7, the *p*-value of 0.0122 for the HAp/BG ratio indicates that this is an important factor in crystal size. The binary interactions between the process factor temperature (C) and the mixture components A (BG) and B (HAp) are also seen to significantly affect HAp/BG composite crystal size with *p*-values of 0.018 and 0.0011, respectively. The ternary interaction with *p*-value 0.6112 does not significantly influence crystal size.

The correlation-coefficient values of a model should be close to 1. Adequate precision, which is the signal/noise ratio, compares the amplitude of predicted values at designed points with a predicted mean error. Values above 4 indicate sufficient model accuracy. From the table, R^2^ = 0.9878, R^2^ adjusted = 0.9756, R^2^ predicted = 0.9601, and adequate precision = 24.103. As is clear, correlation coefficient values (i.e., 0.987) are close to 1 and the value of adequate precision (i.e., 24.103) indicates sufficient model validity.

The equation in terms of real components and actual factors for crystal size in HAp/BG nanocomposites is given in Equation (8).
(8)Crystal size=37.729×BG+36.527×HAp−63.601×BG×HAp+0.00337×BG×Temp+0.0231×HAp×Temp+0.170×BG×HAp×Temp

After removing interactions with *p*-values of 0.05, the final equation for crystal size in HAp/BG nanocomposites is shown in Equation (9).
(9)Crystal size=27.143×BG+34.974×HAp−40.074×BG×HAp+0.00806×BG×Temp+0.0343×HAp×Temp

Figure 6 shows the effect of HAp/BG ratio on the HAp crystal size in the HAp/BG composites. From Table 6 and Figure 6, increasing the BG ratio from 10 to 50 at a constant temperature decreases crystal size. These results are in good agreement with the XRD results shown earlier where it was noted that the existence of the silicon phase in BG inhibits the growth of HAp crystal in the HAp/BG composite. By increasing Si content, the HAp surface was covered by an amorphous phase (i.e., BG). Consequently, the thickness of the amorphous phase increased and Si acted as a barrier to the growth of HAp particle size [27].

By studying the interaction of the two mixture components and the process temperature on the response of the model, 2D contour and 3D response level diagrams were plotted (Figure 7). These diagrams show that increasing component A (BG) from 10% to 50% decreases the crystal size of nanocomposites and suggest that HAp/BG ratio is a primary factor influencing composite crystal size. The same results have been found in a study conducted by [37], which reported that silicon in the BG phase inhibited the HAp crystal growth in HAp/BG composite.

Furthermore, Figure 7a,b also support the finding that, for a fixed HAp/BG ratio, an increase of hydrothermal temperature (from 100 °C to 180 °C) increases the crystal size, as observed in the previous analyses.

### 3.6. Analysis of Variance for HAp Crystallinity in HAp/BG Composite Nanopowder

Table 8 presents important measures of model adequacy (i.e., R^2^, adjusted R^2^, predicted R^2^, and adequate precision). A quadratic model and a linear model were used for the mixture order and process order for the crystallinity response, respectively, because of *p*-value 0.05. From Table 8, the three binary interactions between A (BG) and B (HAp), B (HAp) d C (temperature as process factor), and A and C have *p*-values ≤0.05 and are therefore effective interactions. However, the ternary interaction between ABC with *p*-value 0.4934 does not significantly influence HAp/BG composite crystallinity. From Table 8, R^2^ = 0.9976, predicted R^2^ = 0.97, adjusted R^2^ = 0.9952, and adequate precision = 49.5698, which indicate that the model is sufficiently valid.

The equation for crystallinity HAp/BG nanocomposite in terms of real components and actual factors is shown in Equation (10).
(10)Crystallinity=−29.068×BG+69.543×HAp+108.247×BG×HAp+0.291×BG×Temp+0.0361×HAp×Temp−0.196×BG×HAp×Temp

The final equation after removing interactions with *p*-value 0.05 for crystallinity of HAp/BG nanocomposite in terms of real components and actual factors is shown in Equation (11).
(11)Crystallinity=−16.848×BG+71.335×HAp+81.089×BG×HAp+0.2019×BG×Temp+0.023×HAp×Temp

Combining the results from Table 6 and Figure 8, we also found that an increase in BG from 10% to 50% in the HAp/BG ratio at a fixed hydrothermal temperature decreased the degree of crystallinity. Similar results have been found by the authors of [36,37], who reported that an increase in the BG phase in the HAp/BG structure leads to a decrease in the degree of crystallinity of HAp/BG nanopowders. These studies reported that, compared to pure HAp powder and HAp/BG powder, the silicon group in the BG phase obstructs the crystallization of the HAp phase by limiting atomic arrangement [36,37].

Figure 9a,b show the relationships between the degree of crystallinity, hydrothermal process temperature, and the HAp/BG ratio. As indicated in Figure 8, the degree of crystallinity decreased with higher HAp/BG ratios, suggesting that BG content is an effective factor in the crystallinity of the composites. A study by [36] found that BG content in HAp/BG composites influences crystallinity whereby silicon groups from BG act as a barrier to crystallinity by limiting atomic arrangement [36].

Furthermore, Figure 9 also indicates that increasing hydrothermal temperature from 100 to 180 °C at a constant HAp/BG ratio increases the degrees of crystallinity and that at 180°C, a high degree of crystallinity for HAp in HAp/BG composite was obtained.

As indicated in Figure 10, the values predicted by software and the actual response values for crystal size and crystallinity are very closely correlated [47]. The correlation coefficients of R^2^ = 0.9878 for crystal size and R^2^ = 0.99 for crystallinity confirm a good model fit.

Figure 11 indicates the optimal region. Under optimal conditions (HAp 0.726/BG 0.274, temperature 180 °C), nanocomposites with the highest degree of crystallinity (79.08%) and the smallest size (33.49 nm) were obtained.

From the Design of Experiments with combined mixture–process design and the SEM, EDX, and XRD results, the optimum temperature and HAp/BG ratio for the synthesis of HAp/BG composites with small size, maximum degree of crystallinity, and a Ca/P ratio close to 1.67 are 180 °C and HAp70/BG30, respectively.

### 3.7. In Vitro Behavior of Samples in Simulated Body Fluid (SBF)

To investigate the bioactivity of HAp/BG composite nanopowder, samples were immersed in SBF solution at 37 °C and pH 7.4 for 14 days. Figure 12 shows that an apatite layer precipitated on the HAp/BG composite nanopowder. It is noticeable that adding BG up to 30% into the HAp structure promoted the formation of the apatite layer on the composite nanopowder surface (Figure 12b). SEM micrographs show complete coverage of the HAp70/BG30 composite surface by apatite after 14 days of immersion into SBF solution. This significant increase of bioactivity in this HAp/BG composite can also be attributed to the smaller HAp particle size used in the composite, which increases bioactivity [48]. Initially, Si was not present in the SBF solution and was only released after immersion of HAp/BG samples into it. Hence, the Si released in the SBF solution can only be due to the dissolution of Si available in bioactive glass, making its concentration in SBF a good criterion for studying the degradation behavior of HAp composite with BG. This suggested that the bioactivity of HAp would be improved by the entry of BG into the HAp structure (i.e., the formation of silicate HAp by sintering the HAp composite at 700 °C).

To confirm the formation of an apatite layer on the surface of HAp and HAp/BG nanopowders after immersion in SBF solution for 14 days, we used the FTIR results (Figure 13). From the FTIR spectra (Figure 13), after immersion of samples in SBF solution, different P-O bonds in 600–700 and 900–1100 cm^−1^ wavenumber are sharper than when the sample is not located in SBF solution, as is shown in Figure 13. This indicated that the increased percentage of apatite phase was due to the formation of an apatite layer after immersion of samples in SBF solution. As described before, during sintering HAp/BG samples at 700 °C, part of BG has crystallized and formed the wollastonite phase. Therefore, in the FTIR result, wavenumbers in the range of 798 and 461 cm^−1^ are attributed to Si-O bonds in the glass and wollastonite phase. A similar result was obtained by previous workers [28,49].

Previous studies have shown that both composite dissolution in the SBF solution (i.e., the increase of ion concentration in the SBF solution) and apatite layer deposition (i.e., the reduction of ion concentration in the SBF solution) occur more quickly than in pure HAp and BG. The high solubility of the BG/HAp composite in the SBF solution which causes rapid supersaturation results in the rapid deposition of the apatite layer from the SBF solution [50].

In another study, [37] synthesized silicon-substituted HAp using a hydrothermal method. According to the bioactivity results of that study, the bioactivity of Si-HAp is greater than HAp alone due to lower crystallinity and higher solubility of Si-HAp in SBF solution, which increases supersaturation and leads to high nucleation density [37].

In another study conducted by [28], HAp/BG composite powder with different weight ratios was synthesized and the bioactivity of these composites was evaluated in the SBF solution. Based on the results of SEM imaging and on the apatite layer formed in the SBF solution, it was reported that the HAp70/BG30 composite exhibited higher bioactivity than BG alone. Some of the BG crystallized to form wollastonite phase after composite sintering at 1000 °C and the bioactivity and dissolution rate increased in the presence of silicon. There is considerable evidence that silicon is important for bone formation in bioactive ceramic glasses containing silicate because silanol groups in ceramic glass and BG act as a catalyst for HAp phase budding in forming the apatite surface layer [28,49].

## 4. Conclusions

In this study, HAp/BG composite nanopowder was synthesized via hydrothermal method to evaluate the effect of hydrothermal temperature and component ratio on crystal size and crystallinity. A combined mixture–process design in DOE was used to design a total of 11 experiments. Contour and 3D mixture–process plots demonstrated that the most significant variable influencing crystallinity and crystal size was component ratio. Moreover, the degree of crystallinity of HAp and the size of crystals increased with hydrothermal temperature from 100 to 180 °C. However, the effect of this parameter on the degree of crystallinity was weaker than the component ratio. From the DOE with the combined mixture–process design and from the SEM, EDX, FTIR, and XRD results, the optimum temperature and the best HAp/BG ratio for the synthesis of HAp/BG composite nanopowder with small size, maximum crystallinity, and a Ca/P ratio close to 1.71 (which is close to its stoichiometry value) were 180 °C and HAp70/BG30, respectively. From this study, the addition of BG and the formation of HAp/BG composite nanopowder can be a significant method to improve the bioactivity of hydroxyapatite in SBF solution and it looks very promising for the production of HAp/BG glass composites.

## Figures and Tables

**Figure 1 nanomaterials-12-02264-f001:**
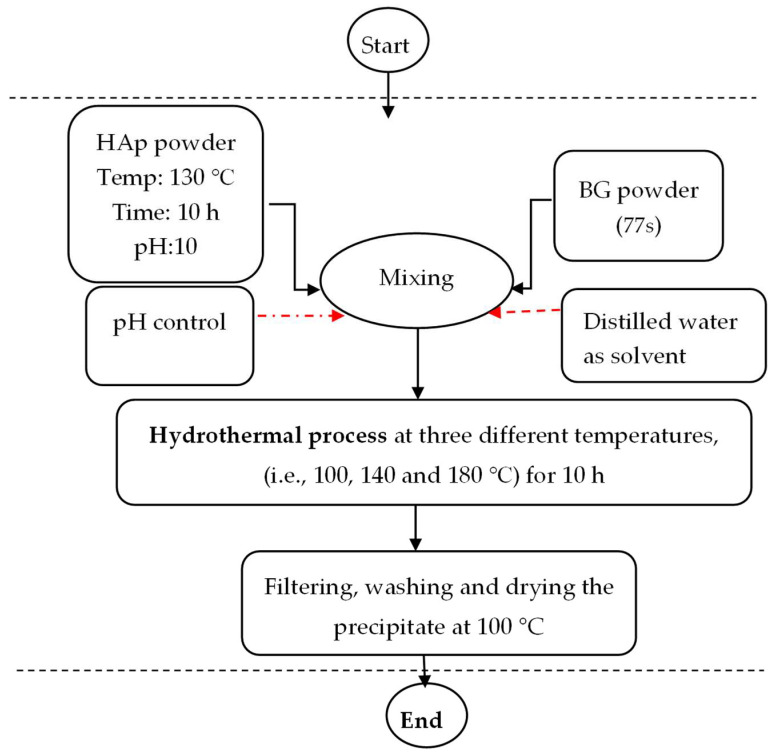
Flowchart for the synthesis of HAp/BG nanocomposite by the hydrothermal method.

**Figure 2 nanomaterials-12-02264-f002:**
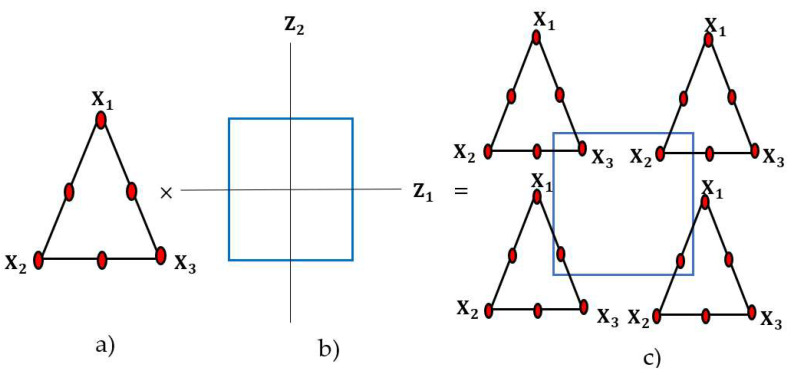
Experimental points in the combined design: (**a**) mixture design for three components; (**b**) factorial design for two process variables; (**c**) combination of mixture components with process variables.

**Figure 3 nanomaterials-12-02264-f003:**
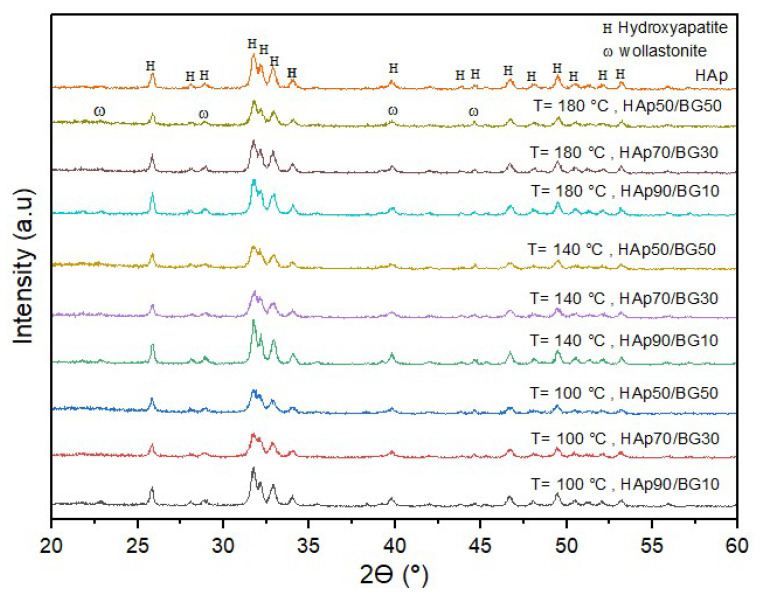
XRD patterns for HAp/BG composites in different conditions based on DOE.

**Figure 4 nanomaterials-12-02264-f004:**
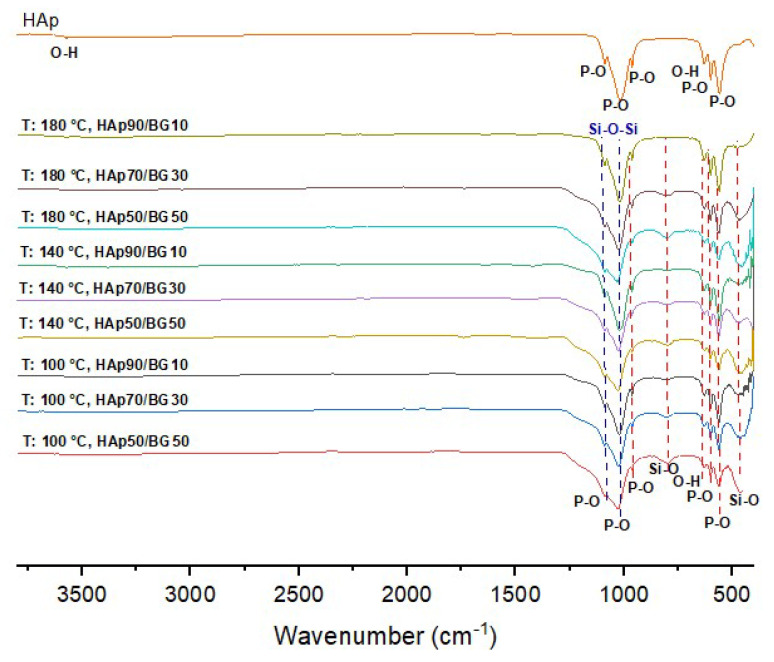
FTIR spectra of HAp and HAp/BG nanocomposites under different conditions based on DOE.

**Figure 5 nanomaterials-12-02264-f005:**
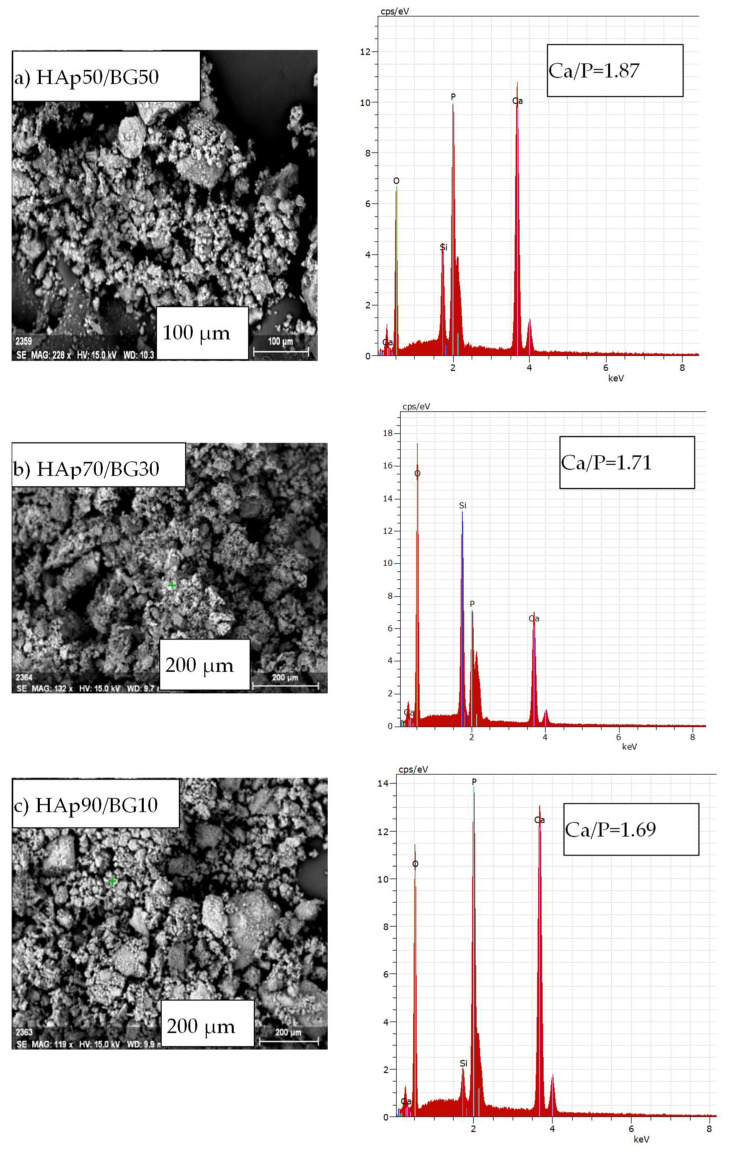
SEM and EDX results of HAp/BG composite nanopowders with three different weight ratios at 180 °C.

**Figure 6 nanomaterials-12-02264-f006:**
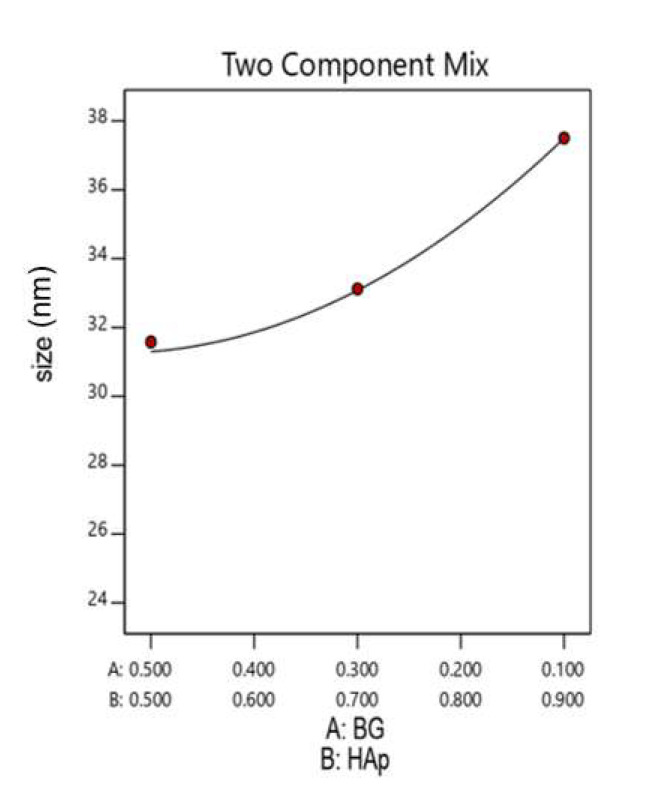
Effect of HAp/BG composite mixture components on HAp crystal size.

**Figure 7 nanomaterials-12-02264-f007:**
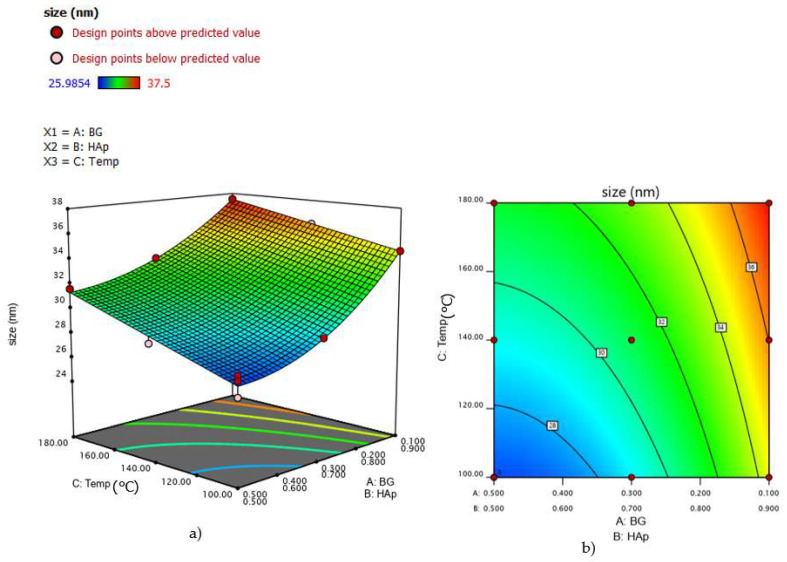
(**a**) 3D mixture–process diagram; (**b**) 2D mixture–process contour plot of process conditions that influence HAp/BG composite crystal size.

**Figure 8 nanomaterials-12-02264-f008:**
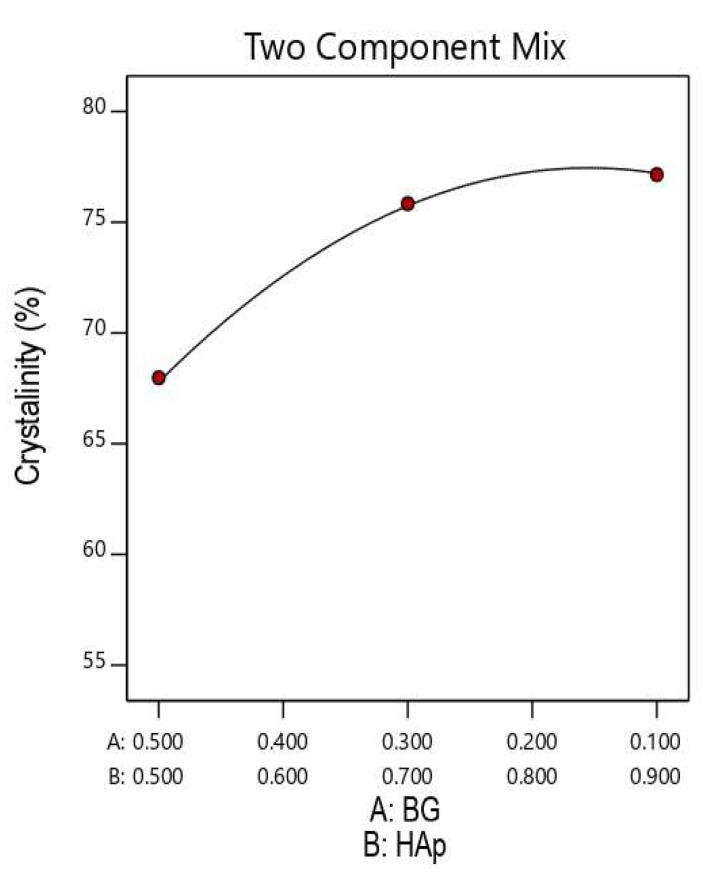
Effect of HAp/BG composite mixture components on HAp crystallinity.

**Figure 9 nanomaterials-12-02264-f009:**
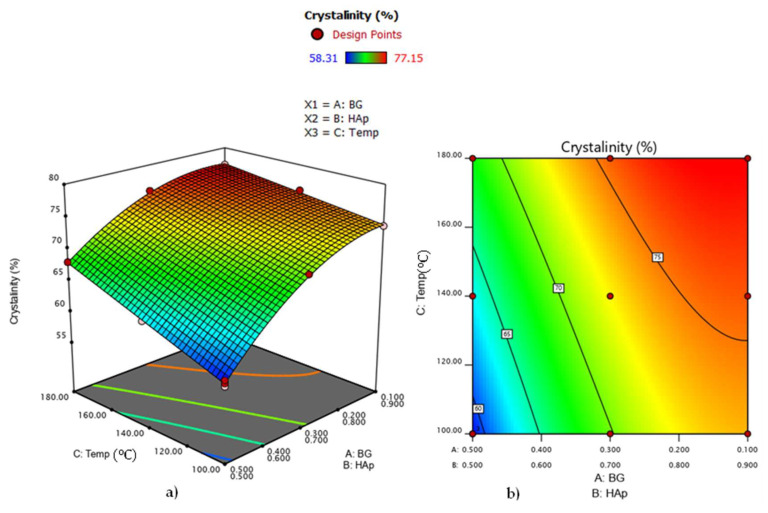
(**a**) 3D mixture–process diagram; (**b**) 2D mixture–process contour plot of process conditions that influence HAp/BG composite crystallinity.

**Figure 10 nanomaterials-12-02264-f010:**
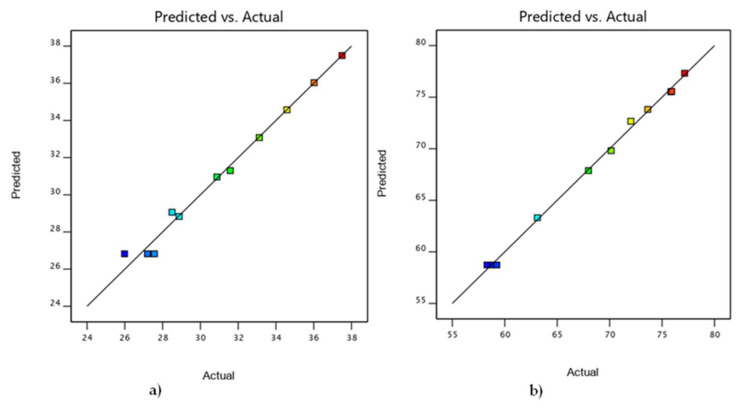
Relation between experimental and model predictions for (**a**) crystal size, (**b**) crystallinity responses.

**Figure 11 nanomaterials-12-02264-f011:**
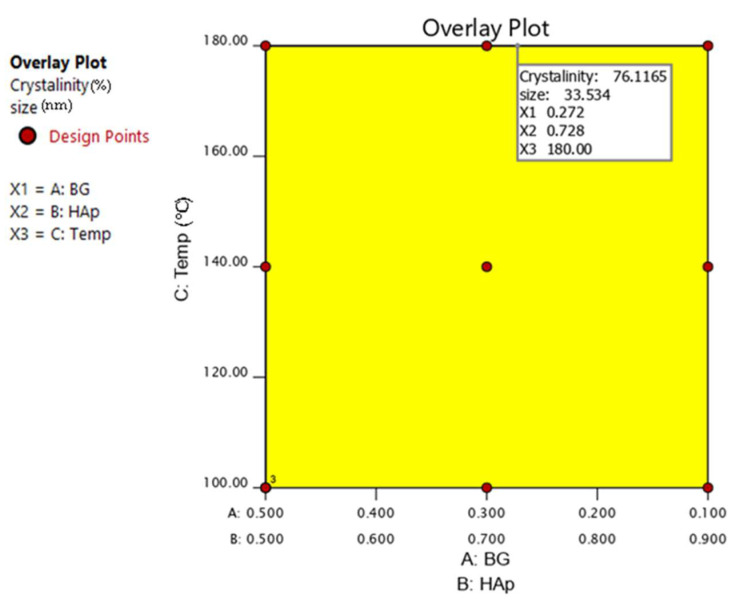
Overlay plot, showing the optimal point of each component and process factor in the composite.

**Figure 12 nanomaterials-12-02264-f012:**
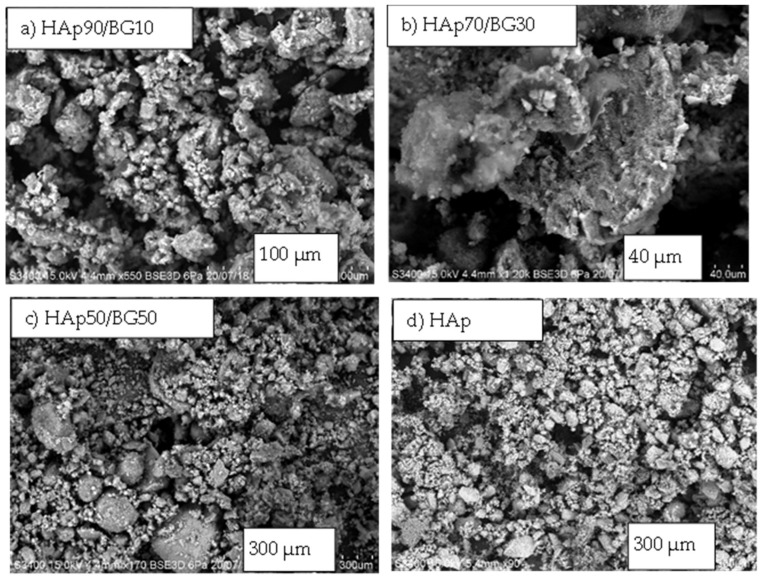
SEM images of HAp and HAp/BG composite nanopowder after immersion in SBF.

**Figure 13 nanomaterials-12-02264-f013:**
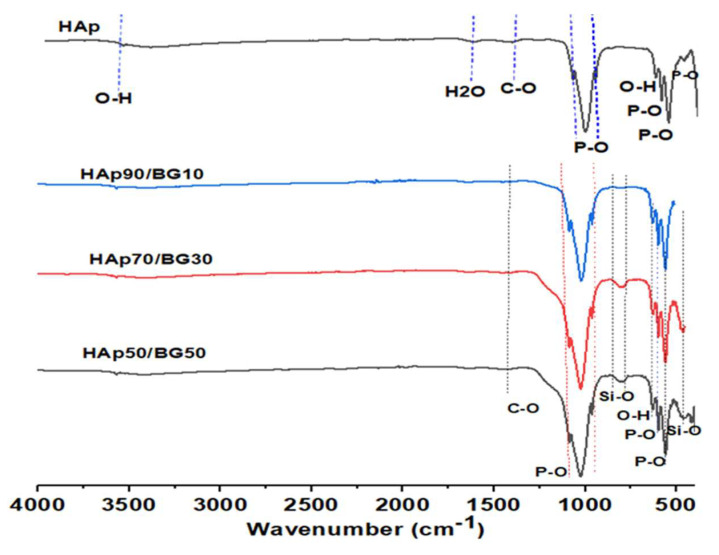
FTIR spectra HAp and HAp/BG composites after immersion in SBF solution.

**Table 1 nanomaterials-12-02264-t001:** Mixture components and their ratios in combined design.

Component	Name	Low	High
A	BG *	10	50
B	HAp	50	90

* Bioglass.

**Table 2 nanomaterials-12-02264-t002:** Numeric factor and its levels in combined design.

Factor	Name	Level 1	Level 2
C	Temperature	100	180

**Table 3 nanomaterials-12-02264-t003:** Combined design and component values.

Run	Component 1 A: BG	Component 2 B: HAp	Factor C: Temp
1	10	90	140
2	50	50	100
3	50	50	180
4	30	70	140
5	50	50	100
6	50	50	140
7	50	50	100
8	10	90	100
9	30	70	100
10	30	70	180
11	10	90	180

**Table 4 nanomaterials-12-02264-t004:** Crystallinity and crystal size of HAp/BG composites at hydrothermal process temperature 100 °C.

Samples	Component 1	Component 2	Process Factor	Responses	
	A: BG	B: HAp	Temperature	Crystallinity	Crystal Size
	wt %	wt %	°C	%	nm
1	10	90	180 °C	77.15	37.5
2	30	70	180 °C	75.84	33.12
3	50	50	180 °C	67.98	31.58
4	10	90	140 °C	75.91	36.02
5	30	70	140 °C	72.02	30.87
6	50	50	140 °C	63.11	28.50
7	10	90	100 °C	73.46	34.58
8	30	70	100 °C	70.14	28.87
9	50	50	100 °C	58.31	27.2

**Table 5 nanomaterials-12-02264-t005:** EDX results of HAp/BG nanocomposites processed at 180 °C.

-	Component 1	Component 2	Responses
Samples	A: BG	B: HAp	Ca/P
	wt %	wt %	%
1	10	90	1.69
2	30	70	1.71
3	50	50	1.87

**Table 6 nanomaterials-12-02264-t006:** DOE predictions vs. values determined by experiment.

-	Component 1	Component 2	Factor 3	Response 1		Response 2	
Run	A:BG	B: HAp	C: Temp	Crystal Size (nm)		Crystallinity (%)	
				Actual	Predicted	Actual	Predicted
1	10	90	140.00	36.02	36.03	75.91	75.57
2	50	50	100.00	27.2	26.82	58.31	58.73
3	50	50	180.00	31.58	31.30	67.98	67.88
4	30	70	140.00	30.8759	30.96	72.02	72.67
5	50	50	100.00	25.9854	26.82	59.24	58.73
6	50	50	140.00	28.501	29.06	63.11	63.31
7	50	50	100.00	27.5647	26.82	58.75	58.73
8	10	90	100.00	34.58	34.57	73.64	73.81
9	30	70	100.00	28.8754	28.84	70.14	69.82
10	30	70	180.00	33.1183	33.08	75.84	75.52
11	10	90	180.00	37.5	37.49	77.15	77.32

**Table 7 nanomaterials-12-02264-t007:** ANOVA for crystal size for quadratic and linear models.

Source	Sum of Squares	df	Mean Square	F-Value	*p*-Value	
Model	145.40	5	29.08	80.94	< 0.0001	significant
Linear Mixture	113.30	1	113.30	315.36	< 0.0001	-
AB	5.28	1	5.28	14.70	0.00122	-
AC	4.26	1	4.26	11.87	0.00183	-
BC	16.03	1	16.03	44.62	0.0011	-
ABC	0.1055	1	0.1055	0.2936	0.6112	-
Residual	1.80	5	0.3593	-	-	-
Lack of Fit	0.429	3	0.143	0.209	0.8833	not significant
Pure Error	1.37	2	0.6837	-	-	-
Cor Total	147.19	10	-	-	-	-

R^2^ = 0.9878, predicted R^2^ = 0.9601, adjusted R^2^ = 0.9756, adequate precision = 24.1030.

**Table 8 nanomaterials-12-02264-t008:** ANOVA table of crystallinity for quadratic and linear model.

**Source**	**Sum of Squares**	**df**	**Mean Square**	**F-Value**	** *p* ** **-Value**	
Model	537.60	5	107.52	417.08	< 0.0001	significant
Linear Mixture	411.45	1	411.45	1596.07	< 0.0001	-
AB	21.77	1	21.77	84.44	0.0003	-
AC	6.16	1	6.16	23.90	0.0045	-
BC	66.94	1	66.94	259.67	0.0001	-
ABC	0.1406	1	0.1406	0.5452	0.4934	-
Residual	1.29	5	0.2578			-
Lack of Fit	0.856	3	0.2854	1.32	0.4587	not significant
Pure Error	0.4329	2	0.2164	-	-	-
Cor Total	538.89	10	-	-	-	-

R^2^ = 0.99, predicted R^2^ = 0.9775, adjusted R^2^ = 0.9952, adequate precision = 49.5698.

## Data Availability

Data is contained within the article.

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
