# Peer review of "Synthesis of Hydroxyapatite/Bioglass Composite Nanopowder Using Design of Experiments"

_nanomaterials, 2022, doi:10.3390/nano12132264_

Round 1

Reviewer 1 Report

I highly assess the proposed approach to optimization of processing the hydroxyapatite - bioglass composite which is the successive step in the development of this bone-substituting biomaterial. Despite that, I have some minor remarks, indicated also by color in the attached reviewed manuscript.

Page 1: "Although most studies on bone scaffolds have used the calcium phosphate family (especially HAp) due to its specific properties, this material has limitations such as low absorption rate by the body and lack of connection to soft tissue. Studies have shown that the placement of bioactive ceramics, such as calcium phosphate and bioactive glasses, in composites yields better efficacy in the development and differentiation of bone cells". Both sentences need appropriate references and some clarification. What "specific properties"mean? Then, HAp is not strictly absorbed by a body, but transforms in human apatite by double change chemical reaction, the process which needs time because of diffusion and chemical reaction rate-limiting steps. Finally, what soft tissue do you think of? It is rather hard tissue, already existing human bone.

Page 1:"limited bone binding resources and other options". Please clarify this phrase, what do you mean by limited resources and other options, now it is too general.

Page 1: "Hence, tissue engineering seeks to restore the biological activity of bones by integrating osteoblast cells and appropriate biocompatible materials. For this purpose, in bone tissue engineering, stem cells are cul-tured on suitable scaffolds under controlled conditions (Hollinger et al., 2004)". This reference is useless as this paper is aimed to another research field without any biological tests.

Page 3: "Hap". Change to HAp.

Page 3: "Ammonium solution NH3". Ammonia in water solution is a compound named ammonium hydroxide.

Page 4: "were used to prepare 77s". Truly, this designation souds strange as it is not related to material composition. But, you may leave it or change it, as you wish.

Lines: 133 (Teflon), 183, 185, 188, 189, 190 (Wollastonite, Calcium silicate). All these compound names shood start with small and not with capital letters.

Page 5: "Debby Scherrer equations (Eq. 4 and Eq. 5, respectively)". Only a single equation is shown.

Page 5: "Kokubo guidelines". Please give either a chemical composition of SBF or a reference to it.

Page 5: "immesrsed". language errors, properly "immersed".

Page 10: "were aggregate". Grammar error, properly "were aggregated".

Page 13: "R2adjusted". a lack of a space between "R2" and "adjusted".

Page 15: "the three binary interactions between A (BG) and B (HAp)
B Error! Bookmark not defined". It must be corrected.

Page 16: "Aminian et al. (2011). who". Delete a point after paretheses.

Author Response

Manuscript ID: nanomaterials-1721377

Title: Synthesis of Hydroxyapatite / Bioglass Composite Nanopowder using Design of Experiments

I highly assess the proposed approach to optimization of processing the hydroxyapatite - bioglass composite which is the successive step in the development of this bone-substituting biomaterial. Despite that, I have some minor remarks, indicated also by colour in the attached reviewed manuscript.

Reviewer #1: Page 1: "Although most studies on bone scaffolds have used the calcium phosphate family (especially HAp) due to its specific properties, this material has limitations such as low absorption rate by the body and lack of connection to soft tissue. Studies have shown that the placement of bioactive ceramics, such as calcium phosphate and bioactive glasses, in composites yields better efficacy in the development and differentiation of bone cells". Both sentences need appropriate references and some clarification. What "specific properties"mean? Then, HAp is not strictly absorbed by a body, but transforms in human apatite by double change chemical reaction, the process which needs time because of diffusion and chemical reaction rate-limiting steps. Finally, what soft tissue do you think of? It is rather hard tissue, already existing human bone.

Page 1:"limited bone binding resources and other options". Please clarify this phrase, what do you mean by limited resources and other options, now it is too general.

Page 1: "Hence, tissue engineering seeks to restore the biological activity of bones by integrating osteoblast cells and appropriate biocompatible materials. For this purpose, in bone tissue engineering, stem cells are cul-tured on suitable scaffolds under controlled conditions (Hollinger et al., 2004)". This reference is useless as this paper is aimed to another research field without any biological tests.

Author response: We appreciate the reviewer suggestion. The manuscript is revised based on these comments. Fist section of introduction rewritten again and some appropriate references are added (page 1, 2). 

Reviewer: Page 3: "Hap". Change to HAp.

Author response: The manuscript is revised based on these comments. Hap is changed to HAp in the whole manuscript.

Reviewer: Page 3: "Ammonium solution NH3". Ammonia in water solution is a compound named ammonium hydroxide.

Author response: The manuscript is revised based on these comments. In the manuscript "Ammonium solution NH3" is changed to NH3. (Page 2)

Reviewer: Page 4: "were used to prepare 77s". Truly, this designation souds strange as it is not related to material composition. But, you may leave it or change it, as you wish.

Author response: The manuscript is revised based on these comments. Synthesis of BG (77s) section is rewritten in a proper way. (Page 3)

Reviewer: Lines: 133 (Teflon), 183, 185, 188, 189, 190 (Wollastonite, Calcium silicate). All these compound names shood start with small and not with capital letters.

Author response: The manuscript is revised based on these comments.  All these compound names are written with small letters and highlighted in the whole manuscript.

Reviewer: Page 5: "Debby Scherrer equations (Eq. 4 and Eq. 5, respectively)". Only a single equation is shown.

Author response: The manuscript is revised based on these comments.  Debby Scherrer equation (i.e. equation 2) and degree of crystallinity equation (i.e. equation 3) are highlighted in the manuscript. (Page 4)

 Reviewer: Page 5: "Kokubo guidelines". Please give either a chemical composition of SBF or a reference to it.

Author response: The manuscript is revised based on these comments.  Two references (I.e. reference No  21 and 22) are added to kokubo guidelines. (Page 5)

Reviewer: Page 5: "immesrsed". language errors, properly "immersed".

Author response: The manuscript is revised based on these comments. In the manuscript "immesrsed" is changed to "immersed" and highlighted. (Page 5)

Reviewer: Page 10: "were aggregate". Grammar error, properly "were aggregated".

Author response: The manuscript is revised based on these comments. In the manuscript the phrase "were aggregate" is changed to "were aggregated". (Page 10)

Reviewer: Page 13: "R2adjusted". a lack of a space between "R2" and "adjusted".

Author response: The manuscript is revised based on these comments. (Page 12)

Reviewer: Page 15: "the three binary interactions between A (BG) and B (HAp)

B Error! Bookmark not defined". It must be corrected.

Author response: The manuscript is revised based on these comments. A (BG) and B (HAp)

B Error! Bookmark not defined" is changed to (A (BG) and B (HAp), B (HAp) d C (temperature as process factor)). (Page 14)

Reviewer: Page 16: "Aminian et al. (2011). who". Delete a point after paretheses.

Author response: The manuscript is revised based on these comments. (Page 15)

Reviewer 2 Report

The paper "Synthesis of Hydroxyapatite / Bioglass Composite Nanopowder using Design of Experiments” by Ebrahimi et al deals with the production of scaffolds made of HA and bioglass nanopowders. The process of production was analysed using a DOE methodology. These composites were analysed by means of XRD, SEM/EDX, immersion tests, FTIR to determine the composition, and bioactivity.

The results are in the scope of Nanomaterials journal. These can be useful, but I find some points to be addressed:

GENERAL COMMENTS:

This work is interesting and well written; however, after reading the manuscript, I would like to point some considerations:

  1. Authors should explain how were selected the high and low levels (Tables 2 and 3) for the design of experiments. It is not clear how these levels have been selected and the origin of this decision.
  2. From the manuscript, it is not clear why is relevant the determination of the crystal size and crystallinity as a function of the processing parameters, i.e. a DOE analysis is performed to optimize these parameters, but why? Please, explain it to the potential reader.
  3. In section 3.7 bioactivity after SBF immersion is studied. Presence of apatite is detected by SEM imaging. I think FTIR results should be included as normally, the presence of apatite is demonstrated by this or a similar technique, not only by SEM images.

PARTICULAR COMMENTS:

  • (Page 1) I would change extracellular networks with extracellular matrix.
  • (Page 2) It is stated that HAp-based nanocomposites have an increased applicability as load-bearing tissue. Bioceramics and bioactive glasses have the problem that mechanical properties are not suitable for load-bearing applications. Composites containing Hap, but not as a matrix, can be suitable for load-bearing, but in general not. Authors should clarify this point.
  • (Pages 2-3) I think point 1.1 would be more suitable for materials and methods section than for the introduction. The introduction should introduce the problem under study, it is not a description of one the methods used in the work.
  • (Page 3) Equation (4) is not mentioned in the main text. Please, introduce it.
  • (Page 4) Some text is hidden in flowchart provided in Fig. 2
  • (Page 5-6) There is no Table 3. Please, renumber all the tables.
  • (Page 5) Please, clarify if the samples were incubated in static or dynamic conditions (i.e. in a orbital incubator)
  • (Page 10) Equation (7) not mentioned in the main text Please, introduce it.
  • (Page 11) In the text is mentioned “(i.e. Carbone)” Do you mean carbon or carbonate?. Please, correct it.
  • (Page 13) Equations for the crystal size after ANOVA were bad numbered (they are not 6 and 7, as there are previously same numbers). I think these eqs should be fixed to put Crystal size=…
  • (Page 14) Which are the units for the crystal size in Table 6?
  • (Page 14) Y axis in Fig. 6 (size and temperature) should include units (same in Fig. 7, 9, 11). Notice that in Fig. 6 the ratio BG/HAp decreases to the right. This is quite contrary to the normal representation of graphs. Normally to the right the ratio should increase, not decrease. I think also, the representation of the ratio, could be clearer if simplified fractions are used in this and other graphs, i.e. instead 0.500/0.500 it would be clearer 1/1 or 1:1).
  • (Page 15) There is a missing reference in Line 342.
  • (Page 16) Equations for the crystallinity after ANOVA were bad numbered (they are not 8 and 9, as there are previously same numbers). I think these eqs should be fixed to put Crystal size=…
  • (Page 19) Chemical reactions involving formation of apatite should be improved. These are not numbered and they are written using plane text and an editor. On the other hand, which is the utility of the inclusion of the steps giving apatite? This can be found in many research articles. Moreover, these steps are included only as something informative, as they are not discussed or related to the current results.

Author Response

Manuscript ID: nanomaterials-1721377

Title: Synthesis of Hydroxyapatite / Bioglass Composite Nanopowder using Design of Experiment

Response to reviewer 2

Reviewer: The paper "Synthesis of Hydroxyapatite / Bioglass Composite Nanopowder using Design of Experiments” by Ebrahimi et al deals with the production of scaffolds made of HA and bioglass nanopowders. The process of production was analysed using a DOE methodology. These composites were analysed by means of XRD, SEM/EDX, immersion tests, FTIR to determine the composition, and bioactivity.

The results are in the scope of Nanomaterials journal. These can be useful, but I find some points to be addressed:

This work is interesting and well written; however, after reading the manuscript, I would like to point some considerations:

  1. Authors should explain how were selected the high and low levels (Tables 2 and 3) for the design of experiments. It is not clear how these levels have been selected and the origin of this decision.

Author response: We appreciate the reviewer suggestion. The manuscript is revised based on these comments. The process factor level and ratios were selected based on literature review and some new references are add. (i.e. references No. 26-34). (Page 6)

Reviewer: 2.       From the manuscript, it is not clear why is relevant the determination of the crystal size and crystallinity as a function of the processing parameters, i.e. a DOE analysis is performed to optimize these parameters, but why? Please, explain it to the potential reader.

Author response: The manuscript is revised based on these comments. The importance of crystallinity and crystal size as response in DOE are explained in the introductory part. (Page 1,2)

Reviewer: 3.       In section 3.7 bioactivity after SBF immersion is studied. Presence of apatite is detected by SEM imaging. I think FTIR results should be included as normally, the presence of apatite is demonstrated by this or a similar technique, not only by SEM images.

Author response: The manuscript is revised based on these comments. To confirm the SEM which is formation of an apatite layer on the surface of HAp and HAp/BG nanopowders after immersion into SBF solution for 14 dayes, FTIR results are also added in the manuscript. (Page 17)

Reviewer: (Page 1) I would change extracellular networks with extracellular matrix.

Author response: The manuscript is revised based on these comments. Extracellular networks is changed to extracellular matrix and highlighted in the manuscript. (Page 1)

Reviewer: (Page 2) It is stated that HAp-based nanocomposites have an increased applicability as load-bearing tissue. Bioceramics and bioactive glasses have the problem that mechanical properties are not suitable for load-bearing applications. Composites containing Hap, but not as a matrix, can be suitable for load-bearing, but in general not. Authors should clarify this point.

Author response: The manuscript is revised based on these comments. The first section of introduction is rewritten in a proper way. (Page 1,2)

Reviewer:  (Pages 2-3) I think point 1.1 would be more suitable for materials and methods section than for the introduction. The introduction should introduce the problem under study, it is not a description of one the methods used in the work.

Author response: The manuscript is revised based on these comments. The introduction part is rewritten in a proper way. (Page 5)

Reviewer: (Page 3) Equation (4) is not mentioned in the main text. Please, introduce it.

Author response: The manuscript is revised based on these comments. These two equations are mentioned in a proper way in the main text. (Page 4)

Reviewer: (Page 4) Some text is hidden in flowchart provided in Fig. 2

Author response: The manuscript is revised based on these comments. This figure was improved. (Page 4)

Reviewer : (Page 5-6) There is no Table 3. Please, renumber all the tables

Author response: The manuscript is revised based on these comments. All the tables are renumbered in the whole manuscript. 

Reviewer: (Page 5) Please, clarify if the samples were incubated in static or dynamic conditions (i.e. in a orbital incubator)

Author response: The manuscript is revised based on these comments. Samples were immersed in the SBF solution and incubated under static condition at 37 °C for 14 days. (Page 5)

Reviewer: Page 10) Equation (7) not mentioned in the main text Please, introduce it.

Author response: The manuscript is revised based on these comments. “According to this process, the chemical formula of Si-HA can be determined as Equation 7”. (Page 9)

Reviewer: (Page 11) In the text is mentioned “(i.e. Carbone)” Do you mean carbon or carbonate?. Please, correct it.

Author response: The manuscript is revised based on these comments. Carbone is changed to Carbon and Carbon is correct. (Page 10)

Reviewer: (Page 13) Equations for the crystal size after ANOVA were bad numbered (they are not 6 and 7, as there are previously same numbers). I think these eqs should be fixed to put Crystal size=…

Author response: The manuscript is revised based on these comments. Equation 8 and 9 are written in a proper way. (Page 12, 13)

Reviewer: Page 14) Which are the units for the crystal size in Table 6?

Author response: The manuscript is revised based on these comments. In table 6 , nm is added as unit for the crystal size. (Page 12)

Reviewer: (Page 14) Y axis in Fig. 6 (size and temperature) should include units (same in Fig. 7, 9, 11). Notice that in Fig. 6 the ratio BG/HAp decreases to the right. This is quite contrary to the normal representation of graphs. Normally to the right the ratio should increase, not decrease. I think also, the representation of the ratio, could be clearer if simplified fractions are used in this and other graphs, i.e. instead 0.500/0.500 it would be clearer 1/1 or 1:1).

Author response: The manuscript is revised based on these comments. Based on the result  (XRD): By increasing Si content, the HAp surface was covered by an amorphous phase (i.e. BG). Consequently, the thickness of the amorphous phase increased and Si acted as a barrier to the growth of HAp particle size [27]. (Page 13)

Reviewer: (Page 15) There is a missing reference in Line 342.

Author response: The manuscript is revised based on these comments. A (BG) and B (HAp)

B Error! Bookmark not defined" is changed to (A (BG) and B (HAp), B (HAp) d C (temperature as process factor)). (Page 14)

Reviewer: (Page 16) Equations for the crystallinity after ANOVA were bad numbered (they are not 8 and 9, as there are previously same numbers). I think these eqs should be fixed to put Crystal size=…

Author response: The manuscript is revised based on these comments. The equations are renumbered in the whole manuscript.

Reviewer: Page 19) Chemical reactions involving formation of apatite should be improved. These are not numbered and they are written using plane text and an editor. On the other hand, which is the utility of the inclusion of the steps giving apatite? This can be found in many research articles. Moreover, these steps are included only as something informative, as they are not discussed or related to the current results.

Author response: The manuscript is revised based on these comments. This part is removed from the manuscript.

Reviewer 3 Report

The introduction should be dedicated to present critical analysis of state-of-the-art related work to justify the objective of the study. In this case, overall, the introduction section is simple without a comprehensive description. Therefore, the introduction should be revised by highlighting whether previous studies have carried out, what is the weakness of previous studies and why this study is meaningful and necessary. The introduction has deficiency citation to valuable works published recently. Section 1.1 should be divided between the intrusion and Materials and methods section, while keeping only the relevant information for a research article. Thus, the authors are advised to improve and rewrite some portion of the introduction. 

Abstract and Conclusions – please refer to all characterization methods used and relevant results obtained.

At the end of the introduction, the authors should describe the purpose (objectives) of the study, mentioning also the methods used, obviously briefly, but this should be explained in more detail in the methods.

Please improve the scientific rigor of the manuscript. (1) the quality of Figures 2 and 5 should be improved, (2) the equation corresponding to the line 442, (3) line 158, please use the appropriate reference for Kokubo guidelines, (4) the y scale is missing in Figure 2, (5) chemical formula – “SiO2” should be replace by “SiO2”, (6) the scale bar of Figures 5 and 12 is not visible, (7) the XRD files are missing, (8) Table 7 – the repetitive “oC”makes no sense, (8) line 214 – similar results… should be elaborated, (9) Tables 1 makes no sense, (10) expressions such as “same, similar results”makes no sense, etc.

Also, more attention should be paid to the format of the manuscript. Example: the authors did not follow basic rules of MDPI about preparation of manuscript, since the citations in the text are not given in adequate way. According to the MDPI rules references in the text should be given numerically but not by the surname of the first authors. Author Contributions is missing.

Summarizing, the manuscript may be interesting but it needs significantly improvements before it will be accepted to print in this journal. If the manuscript will not be considerable improved, I will not recommend its publication.

Author Response

Manuscript ID: nanomaterials-1721377

Title: Synthesis of Hydroxyapatite / Bioglass Composite Nanopowder using Design of Experiments

Response to reviewer 3

Reviewer: The introduction should be dedicated to present critical analysis of state-of-the-art related work to justify the objective of the study. In this case, overall, the introduction section is simple without a comprehensive description. Therefore, the introduction should be revised by highlighting whether previous studies have carried out, what is the weakness of previous studies and why this study is meaningful and necessary. The introduction has deficiency citation to valuable works published recently. Section 1.1 should be divided between the intrusion and Materials and methods section, while keeping only the relevant information for a research article. Thus, the authors are advised to improve and rewrite some portion of the introduction.

Author response: We appreciate the reviewer suggestion. The manuscript is revised based on these comments. The introduction was rewritten in a proper way. Section 1.1 was shifted to material and method part. (Page 5)

Reviewer: Abstract and Conclusions – please refer to all characterization methods used and relevant results obtained.

Author response: The manuscript is revised based on these comments.

Reviewer: At the end of the introduction, the authors should describe the purpose (objectives) of the study, mentioning also the methods used, obviously briefly, but this should be explained in more detail in the methods.

Author response: The manuscript is revised based on these comments. The objective of study is mentioned at the end of introduction. (Page 2)

Reviewer: the quality of Figures 2 and 5 should be improved.

Author response: The manuscript is revised based on these comments. The quality of Figures 1 and 5 was improved. (page 4, 11)

Reviewer: the equation corresponding to the line 442

Author response: The manuscript is revised based on these comments. Based on other reviewer comments this part was removed from the manuscript.

Reviewer: line 158, please use the appropriate reference for Kokubo guidelines.

Author response: The manuscript is revised based on these comments. Two references (I.e. reference No.  21 and 22) are added to kokubo guidelines. (Page 5)

Reviewer: the y scale is missing in Figure 2.

Author response: The manuscript is revised based on these comments.

Reviewer: chemical formula – “SiO2” should be replace by “SiO2”.

Author response: The manuscript is revised based on these comments. (Page 3)

Reviewer: the scale bar of Figures 5 and 12 is not visible.

Author response: The manuscript is revised based on these comments. The scale bar was added in Figure 5 and 12. (Page 11, 18)

Reviewer: the XRD files are missing.

Author response: The manuscript is revised based on these comments. (Page 8)

Reviewer: Table 7 – the repetitive “oC”makes no sense.

Author response: The manuscript is revised based on these comments.

Reviewer:  line 214 – similar results… should be elaborated.

Author response: The manuscript is revised based on these comments. This line was elaborated to (The paper [37] studies about incorporation silicon into the formulation also showed that decreased crystallinity and crystal size of HAp in the HAp/BG composite.). (Page 8)

Reviewer: Tables 1 makes no sense.

Author response: The manuscript is revised based on these comments. Table 1 was removed and Synthesis of BG (77s) part was rewritten. (Page 3)

Reviewer: expressions such as “same, similar results”makes no sense, etc.

Author response: The manuscript is revised based on these comments. These expression were elaborate in the whole manuscript. (Page 8, 9)

Reviewer: Also, more attention should be paid to the format of the manuscript. Example: the authors did not follow basic rules of MDPI about preparation of manuscript, since the citations in the text are not given in adequate way. According to the MDPI rules references in the text should be given numerically but not by the surname of the first authors. Author Contributions is missing.

Author response: The manuscript is revised based on these comments. The manuscript was rewritten based on MDPI format. The references were given numerically in the whole manuscript.

Reviewer: Summarizing, the manuscript may be interesting but it needs significantly improvements before it will be accepted to print in this journal. If the manuscript will not be considerable improved, I will not recommend its publication.

Author response: We appreciate the reviewer suggestion. We are uploading (a) our point-by-point response to the comments (response to reviewer), (b) an updated manuscript with yellow highlighting indicating changes.

Round 2

Reviewer 2 Report

In the current state, the manuscript entitled "Synthesis of Hydroxyapatite / Bioglass Composite Nanopowder using Design of Experiments" by Ebrahimi et al has been revised according to the comments made by the reviewers. Congratulations to the authors for this nice work.

Author Response

Manuscript ID: nanomaterials-1721377

Title: Synthesis of Hydroxyapatite / Bioglass Composite Nanopowder using Design of Experiments

To: Dear Ms. Maria

Re: Response to reviewer 2

Dear Editor of nanomaterials

Thank you for allowing a resubmission of our manuscript, with an opportunity to address the reviewer comment.

We are uploading (a) our point-by-point response to the comments (below) (response to reviewer), (b) an updated manuscript with yellow highlighting indicating changes, and (c) a clean updated manuscript without highlights (PDF main document).

Best regards,

Prof. Coswald Stephen Sipaut

Reviewer: in the current state, the manuscript entitled "Synthesis of Hydroxyapatite / Bioglass Composite Nanopowder using Design of Experiments" by Ebrahimi et al has been revised according to the comments made by the reviewers. Congratulations to the authors for this nice work.

Author response: We appreciate the reviewer’s comment.

Reviewer 3 Report

Thanks for the clarifications and edits to the manuscript. I recommend the publication with some minor revisions: (a) please detail the title of figure 11; (b) references must follow the journal guidelines/ please check and revise them accordingly.

Author Response

Manuscript ID: nanomaterials-1721377

Title: Synthesis of Hydroxyapatite / Bioglass Composite Nanopowder using Design of Experiments

To: Dear Ms. Maria

Re: Response to reviewer 3

Dear Editor of nanomaterials

Thank you for allowing a resubmission of our manuscript, with an opportunity to address the reviewer comment.

We are uploading (a) our point-by-point response to the comments (below) (response to reviewer), (b) an updated manuscript with yellow highlighting indicating changes, and (c) a clean updated manuscript without highlights (PDF main document).

Best regards,

Prof. Coswald Stephen Sipaut

Reviewer: Thanks for the clarifications and edits to the manuscript. I recommend the publication with some minor revisions: (a) please detail the title of figure 11.

Author response: We appreciate the reviewer suggestion. The manuscript is revised based on these comments. Detail was added in title of the Figure 11. (page 17)

Reviewer: references must follow the journal guidelines/ please check and revise them accordingly.

Author response: The manuscript is revised based on these comments. The references and list references are rewritten based on MDPI-reference- guideline.
